


# Frazil-ice growth rate and dynamics in mixed layers and sub-ice-shelf plumes

David W. Rees Jones[1,2] and Andrew J. Wells[1]

[1]Atmospheric, Oceanic and Planetary Physics, Department of Physics, University of Oxford, Clarendon Laboratory, Parks Road, Oxford, OX1 3PU, UK.
[2]Department of Earth Sciences, University of Oxford, South Parks Road, Oxford, OX1 3AN, UK.

*Correspondence to:* David W. Rees Jones (David.ReesJones@earth.ox.ac.uk)

**Abstract.** The growth of frazil or granular ice is an important mode of ice formation in the cryosphere. Recent advances have improved our understanding of the microphysical processes that control the rate of ice-crystal growth when water is cooled beneath its freezing temperature. These advances suggest that crystals grow much faster than previously thought. In this paper, we consider models of a population of ice crystals with different sizes to provide insight into the treatment of frazil
ice in large-scale models. We consider the role of crystal growth alongside the other physical processes that determine the dynamics of frazil ice. We apply our model to a simple mixed layer (such as at the surface of the ocean) and to a buoyant plume under a floating ice shelf. We provide numerical calculations and scaling arguments to predict the occurrence of frazil-ice explosions, which we show are controlled by crystal growth, nucleation and, gravitational removal. Faster crystal growth, higher secondary nucleation and slower gravitational removal make frazil-ice explosions more likely. We identify steady-state
crystal size distributions, which are largely insensitive to crystal growth rate but are affected by the relative importance of secondary nucleation to gravitational removal. Finally, we show that the fate of plumes underneath ice shelves is dramatically affected by frazil-ice dynamics. Differences in the parameterization of crystal growth and nucleation give rise to radically different predictions of basal accretion and plume dynamics; and can even impact whether a plume reaches the end of the ice shelf or intrudes at depth.

## 1 Introduction

### 1.1 Frazil ice in the environment

Frazil-ice formation is an extremely rapid mode of ice growth occurring as the initial phase of ice growth in a turbulent waters. Frazil ice forms as a suspension of crystals in oceans, lakes, rivers and sub-glacial ice streams from liquid water supercooled beneath its freezing temperature (Martin and Kauffman, 1981; Lawson et al., 1998). Supercooled water at the surface of the
ocean occurs when it cooled efficiently by the atmosphere. Such conditions can occur in gaps in the ice pack (called leads) and in extensive areas of open water (called polynyas), as observed by Skogseth et al. (2009). In some Antarctic regions, frazil ice growth in supercooled water also contributes to the accretion of platelet ice on the underside of sea ice (e.g. Gough et al., 2012; Langhorne et al., 2015).



Frazil ice can also form underneath floating ice shelves at the margins of the Antarctic continent. Plumes of relatively fresh, buoyant 'ice shelf water' (ISW) flow along the underside of the ice shelves. These rise over a depth range of about a kilometre, a range associated with significant variation of the pressure-dependent freezing temperature of seawater, which varies by $-0.76\,°$C/km with depth (Millero and Leung, 1976). Consequently, the temperature of a rising plume can fall beneath the *in situ* freezing temperature (Lewis and Perkin, 1986), triggering the formation of frazil ice. Some of the ice precipitates onto the base of the ice shelf, where it forms ice with a granular texture that can be observed by drilling boreholes through the ice shelf (Engelhardt and Determann, 1987). Frazil-ice formation can affect the dynamics of these plumes by changing their buoyancy directly (because ice is less dense than water), and by changing their temperature and salinity.

It is just becoming possible to assess the role of frazil-ice formation on sea ice and ocean conditions through large scale models (e.g. Galton-Fenzi et al., 2012; Wilchinsky et al., 2015; Smedsrud and Martin, 2015). Such models rely on previous theoretical work concerning frazil-ice dynamics, which was pioneered by Daly (1984). Models of frazil-ice dynamics have been applied to the study of frazil in the upper ocean (Svensson and Omstedt, 1994, 1998), and also to the study of frazil ice beneath ice shelves (Jenkins and Bombosch, 1995; Khazendar and Jenkins, 2003; Smedsrud and Jenkins, 2004; Holland and Feltham, 2005; Jordan et al., 2014, 2015). The theory of frazil-ice dynamics involves parameterizations of a number of physical processes that affect the evolution of a population of ice crystals. In this paper, we revisit the theory of frazil-ice dynamics taking into account new understanding of the microphysics of crystal growth (Rees Jones and Wells, 2015), before suggesting likely implications for these large scale models.

## 1.2 Crystal growth rate

In a recent paper, Rees Jones and Wells (2015) presented numerical evidence that the growth rate of ice crystals has been significantly underestimated in some previous studies as detailed below. In this section, we briefly review this finding, and explain the underlying physical ideas.

Frazil ice is observed to consist of roughly disk-shaped crystals that typically have a much greater radius $R$ than thickness $H$ (McFarlane et al., 2014). Crystal growth is predominantly radial, with attachment kinetics limiting growth in the basal plane and maintaining the disk-shaped geometry for crystals of modest size (Fujioka and Sekerka, 1974). The radial growth rate $G$ of a frazil crystal depends on the rate at which the latent heat released by crystal growth is transported away from the crystal. In general, the radial growth rate can be written in the form

$$\rho_i L G = (\mathrm{Nu}\, k_l \Delta T/H)f, \tag{1}$$

where $\rho_i$ is the density of ice, $L$ is the latent heat of solidification, $\mathrm{Nu}$ is the crystal Nusselt number which equals 1 for purely diffusive growth and can be enhanced by flow, $k_l$ is the thermal conductivity of the liquid phase, $\Delta T$ is the amount of supercooling below the in-situ freezing temperature, and $f$ is a dimensionless geometric factor. A helpful way to interpret equation (1) is to rearrange it into an expression for the rate of crystal-mass growth, which the ice-crystal mass $M = \rho_i \pi R^2 H$. We find

$$L\frac{dM}{dt} = \mathrm{Nu} k_l \Delta T 2\pi R f \propto \mathrm{Nu} k_l \Delta T \frac{A}{\delta}f, \tag{2}$$





The right-hand side is the product of the area for heat transfer $A$ and the heat flux scale $k_l \Delta T / \delta$, where $\delta$ is a thermal boundary layer thickness. Numerical calculations show that $\delta \propto H$ near the crystal edges, which have an area $A \propto RH$. However, $\delta \propto R$ near the crystal faces, which have an area $A \propto R^2$. In either case, the ratio $A/\delta \propto R$. Thus the scaling argument suggests $f \propto 1$ (*cf.* equation 2). It is interesting to note that the mass growth rate of spherical crystals is also proportional to crystal radius $R$,
so the rate of latent heat release seems to depend on crystal size $R$ but not on the details of the geometry.

We now consider three possible parameterizations of crystal growth, which we denote $f_{1,2,3}$. Some previous studies are consistent with the scaling $f \sim 1$. For example, Svensson and Omstedt (1994) and Jenkins and Bombosch (1995) take $f_2 = 1$. By contrast, some later studies are inconsistent with the scaling argument. For example, Smedsrud and Jenkins (2004), Holland et al. (2007) and Galton-Fenzi et al. (2012) take $A \propto RH$ and $\delta \propto R$, which gives a growth rate proportional to $f_3 \equiv H/R \ll 1$, i.e. a very much smaller growth rate. A further complication arises in that it is sometimes additionally assumed that the crystal aspect ratio $h = H/2R$ is constant, rather than the crystal thickness $H$ being constant. In this case, $f_3 \equiv 2h$, which is a constant, like $f_2$, but very much smaller (e.g. Smedsrud and Jenkins (2004) take $h = 0.02$). These papers are illustrative of a wider range of studies (e.g. Svensson and Omstedt, 1998; Khazendar and Jenkins, 2003; Holland and Feltham, 2005; Jordan et al., 2014, 2015; Wilchinsky et al., 2015; Smedsrud and Martin, 2015); recently it appears that growth law $f_3$ has been used most commonly, if not exclusively.

Numerical calculations of the heat transfer by diffusion from a disk-shaped crystal (Rees Jones and Wells, 2015) show that the growth rate depends logarithmically on aspect ratio $f_1(h = H/2R) = 1/[0.9008 - 0.2634 \log(h)]$. This is a weak dependence on aspect ratio: $f_1$ is typically close to $f_2$; however, $f_1$ is some 10–100 times greater than $f_3$, as illustrated in Figure 1.

The presence of salt in seawater reduces the crystal growth rate because the supercooling is reduced and salt rejected by the growing crystal needs to diffuse away. Numerical calculations performed to investigate these effects (Rees Jones and Wells, 2015) support the scaling argument used to account for the effect of salt by Galton-Fenzi et al. (2012), which in turn was based on Holland and Jenkins (1999). For practical modelling purposes, the supercooling needs to be adjusted for the salt content of seawater, and the Nusselt number should be reduced to account for salt diffusion. A good approximation is $\mathrm{Nu} = [1 + 1.4 \times (-aSk_l)/(D_S \rho_i L)]^{-1}$, where $a < 0$ is the rate of change of freezing temperature with salinity $S$, and $D_S$ the diffusivity of salt.

## 2 Frazil-ice dynamics

### 2.1 Physical processes

How does the growth rate of an ice crystal affect the overall ice production rate of a system? To address this question we need to investigate 'frazil-ice dynamics'. We follow the comprehensive framework of the influential reviews of Daly (1984, 1994), which accounts for the evolution of a crystal size distribution in time, in space, and in crystal size space. The evolution occurs through crystal nucleation, growth, flocculation, breakup, and transport by fluid motion. There is a high degree of uncertainty





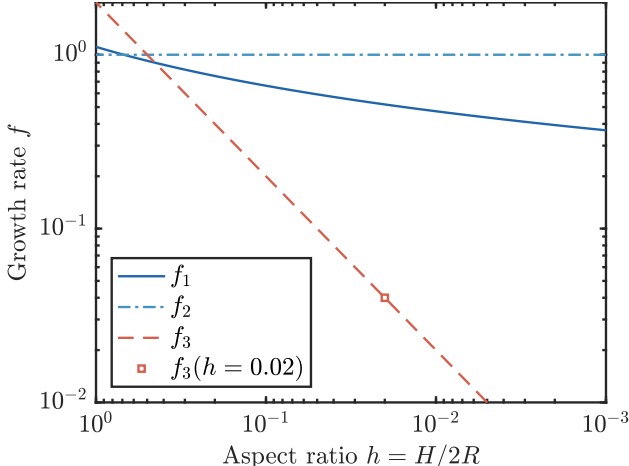

**Figure 1.** Three parameterizations of crystal growth. The parameterization $f_1$ (solid dark blue curve) is the result of a detailed numerical calculation (Rees Jones and Wells, 2015). Parameterizations $f_2$ (dot-dash light blue curve) and $f_3$ (dashed red line) are obtained by scaling analysis, as described in the text. The growth rates $f_1$ and $f_2$ are comparable, but $f_3$ is much smaller at typical small aspect ratios. We also indicate (square marker) the growth rate if a constant aspect ratio $h = 0.02$ is assumed.

in the rate of each of these processes, which in turn drives uncertainty in predictions of crucial, environmentally relevant quantities, such as the total ice production rate.

## 2.2 Mathematical description

In this section we set out continuum equations that describe the evolution of frazil ice in a general framework that can be

applied to a wide range of specific situations, before later focussing on examples of ice growth in a mixed layer, and under ice shelves. Suppose that the size of a crystal can be characterised by a single length scale $R$, the radius of a disk-shaped crystal. We introduce the crystal number density $n$, which is defined as the number of crystals per unit volume of mixture per unit length in crystal-size space. Other quantities can be derived from $n$. For example, the crystal concentration density $c = nV$, where $V = \pi R^2 H$ is the volume of a disk shaped crystal of thickness $H$, and the total crystal concentration $C = \int_0^\infty c \, dR$. The

total number density $N = \int_0^\infty n \, dR$. The density $n$ is a function of time $t$, position $\boldsymbol{x}$ and crystal size $R$, and is governed by (cf. Daly, 1984)

$$\frac{\partial n}{\partial t} + \nabla \cdot (\boldsymbol{u}n) - \nabla(D_c \nabla n) = \tag{3}$$
$$-\frac{\partial}{\partial R}(Gn) - W\frac{\partial n}{\partial z} - \frac{1}{V}\frac{\partial}{\partial R}(BVn) + \dot{N}\delta(R),$$

where $\boldsymbol{u}$ is the fluid velocity and $D_c$ is the crystal diffusivity. For turbulent flows, one approach is to parameterize the effects

of fluid flow $\boldsymbol{u}$ as an enhanced diffusivity, sometimes called a turbulent or eddy diffusivity. The terms on the right-hand side represent frazil dynamics terms, on which we elaborate below.

The first term represents crystal growth, where $G$ is the radial crystal growth rate discussed in section 1.2. For compactness, we rewrite equation (1) as

$$G = G_0 f, \tag{4}$$

where $G_0 = \mathrm{Nu}\,k_l \Delta T/(\rho_i L H)$, and $f$ is given by one of the three growth laws. The effect of this term is to shift the crystal size distribution to larger radii $R$, without increasing the total number of crystals. Thus growth increases crystal concentration, but not the number of crystals.

The second term represents removal due to buoyant crystal rise, where $W$ is an effective crystal rise speed. It is well established that larger crystals rise faster. Recent experimental observations and the parameterization of crystal rise speed are discussed in McFarlane et al. (2014). For the simplest treatment, we use a linear relationship

$$W = W_0 R, \tag{5}$$

with $W_0 = 16\,\mathrm{s}^{-1}$, because more complicated parameterizations do not fit the data much better than this simple fit. Indeed, such a relationship is consistent with the crystal rise being a Stokes settling velocity under the assumption that crystal thickness is constant. The drag is proportional to $\mu W R$, and the buoyancy is proportional to $\Delta \rho g R^2 H$, where $\Delta \rho$ is the density difference between ice and water. Thus balancing drag and buoyancy yields $W \propto R$.

The third term represents the net effect of the processes of flocculation and break up, where $B$ is the rate. Positive $B$ corresponds to flocculation greater than break up. Note that this term is constructed to conserve crystal volume, which is physically appropriate. To see this, multiply equation (3) by $V$ and integrate from $R = 0$ to $R = \infty$. The total volume of ice is unaffected by the flocculation term. To our knowledge, this term has received relatively little attention within the frazil-ice literature. One exception, Svensson and Omstedt (1994) include it and take

$$B = B_0 R^2. \tag{6}$$

As a technical aside, we note that Svensson and Omstedt (1994) describe their flocculation law as linear. However, this linearity applies only to the particular discrete set of equations they present, which use logarithmically spaced size classes. At the continuum level, the quadratic equation (6) applies. There is no direct evidence for the form of this relationship, although Svensson and Omstedt (1994) found it helpful in fitting some experimental data. Their choice matches the intuition that larger crystals might flocculate more readily since they are more likely to come into near contact with other crystals. However, it does not account for the fact that flocculation should increase with frazil concentration. A fuller treatment would take $B$ as an integral of an interaction kernel $K$ multiplied by number density over crystal radius, $B = \int_0^\infty K n \, dR$. This kind of approach has proved fruitful in the theory of sea-ice thickness and floe-size distributions (Thorndike, 2000; Godlovitch et al., 2011; Horvat and Tziperman, 2015; Toppaladoddi and Wettlaufer, 2015). In view of the considerable uncertainties in parameterizing flocculation, we neglect this process in all of our calculations ($B = 0$).

The fourth term represents crystal nucleation, where $\dot{N}$ is nucleation rate. We use the mathematical construct of a delta function $\delta(R)$ in equation (3) to express the fact that nucleated crystals are extremely small. By integrating equation (3) from





$R = 0$ to $R = \epsilon > 0$, it can be shown that the nucleation flux balances the growth of small crystals

$$\lim_{\epsilon \to 0^+} Gn|_{R=\epsilon} = \dot{N}, \tag{7}$$

since the other terms give rise to contributions that are proportional to $\epsilon$ and vanish in the limit $\epsilon \to 0^+$. After some primary nucleation event, nucleation is assumed to be dominated by secondary nucleation, sometimes called collision breeding. Indeed, Daly (1984) argues that homogenous and heterogenous nucleation are extremely unlikely to occur in natural systems because the levels of supercooling achieved are less than $1°\text{C}$. We follow e.g. Svensson and Omstedt (1994) and suppose that collisions between crystals cause microscopic pieces of ice to break off which in turn become new crystals with very small radius. The total nucleation rate depends on the the rate at which a volume is swept out by a crystal and the crystal number density. We write

$$\dot{N} = \tilde{n} \int_0^\infty \pi R^2 U_r n(R) \, dR, \tag{8}$$

where

$$U_r = \sqrt{4\epsilon R^2/15\nu + (W_0 R)^2} \equiv U_0 R \tag{9}$$

is an effective collisional velocity scale taken to be the geometric mean of a velocity scale based on turbulent motions ($\epsilon$ is the turbulent dissipation rate, $\nu$ is the kinematic viscosity) and one based on buoyant crystal rise. We define $U_0 = \sqrt{4\epsilon/15\nu + W_0^2}$, and use a value $\nu = 2 \times 10^{-6} \text{ m}^2\text{ s}^{-1}$. The nucleation efficiency scale $\tilde{n} = \min(N, \tilde{n}_{\max})$, where $\tilde{n}_{\max}$ is a callibration parameter that limits the efficiency of secondary nucleation. Smedsrud (2002) points out that some of the nucleated crystals will be below the so-called 'critical size' for crystals to grow, so it is plausible that $\tilde{n} < N$, but it must be conceded that this parameterization is rather *ad hoc*. We use this formulation primarily for consistency with previous studies, to allow us to isolate the effect of crystal growth rate. It is simply a continuous version of that used by e.g. Svensson and Omstedt (1994); Smedsrud (2002); Smedsrud and Jenkins (2004); Holland and Feltham (2005). Before the efficiency cap is reached, secondary nucleation is a quadratic in the number of crystals, leading to very rapid growth in crystal number.

### 2.3 Numerical methods to solve governing equations

Equation (3) can be discretised in radial space to facilitate numerical solution, following e.g. Svensson and Omstedt (1994). The spatial problem is a standard advection–diffusion problem so we do not discuss here how to discretize the left-hand side of the equation (3) and focus on the crystal interaction terms on the right-hand side. Let $R_i$ be a discrete set of points in radial space, where $1 \leq i \leq M$. We introduce the notation $W_i = W(R_i)$, $G_i = G(R_i)$ and $V_i = V(R_i) = \pi R_i^2 H$. We work in terms of the total number of particles in size class $i$, denoted $m_i$, which evolves according to

$$\frac{\partial m_i}{\partial t} = -\Gamma_i m_i + \Gamma_{i-1} m_{i-1} - W_i \frac{\partial m_i}{\partial z} - \alpha_i m_i \quad (i \geq 2) \tag{10}$$

$$\frac{\partial m_i}{\partial t} = -\Gamma_i m_i - W_i \frac{\partial m_i}{\partial z} + \sum_{j=2}^{j=M} \tilde{n} \pi R_j^2 U_r(R_j) \quad (i = 1)$$





where

$$\Gamma_i = \frac{G_i 2\pi R_i H}{V_{i+1} - V_i}, \tag{11}$$

$$\alpha_i = \tilde{n}\pi R_i^2 U_r(R_i)\frac{V_1}{V_i}, \qquad (i \geq 2) \tag{12}$$

The discrete distribution $n(R_i)$ can be recovered $n_i = m_i/\Delta R_i$, where $\Delta R_i = R_{i+1} - R_i$. We note that equation (10) is only a

first-order discretization in radial space, so an alternative approach could be to use a second order discretization. This numerical representation is conservative, and we use a formulation of secondary nucleation (in terms of $\alpha_i$) that conserves crystal volume even when $V_1$ is non-zero. Note that in the limit $R_1 \rightarrow 0$ and $\Delta R_i \rightarrow 0$ we recover the continuum equations discussed above.

Equation (10) is equivalent to equation (1) in Svensson and Omstedt (1994). They demonstrate that this model is capable of reproducing the main features of the laboratory experiments of Michel (1963) and Carstens (1966), so we do not include

any experimental comparison here. However, we discuss (section 3.3) how such consistency is insufficient to fully validate the model. For practical purposes, we find it advantageous to use a logarithmically spaced set of crystal sizes and test the accuracy of our discretization by increasing the number of size classes to 1024. We find that good accuracy can be achieved with 128 classes, but accuracy noticeably degrades beneath this (*cf.* Holland and Feltham, 2005). Software code to reproduce the calculations in the paper is available (Rees Jones, 2017).

## 3   Frazil ice in a mixed layer

### 3.1   Simplified governing equations

The upper layer of a lake or ocean can sometimes be approximated as a well mixed layer, an approximation that can also be applied to the laboratory experiments of Michel (1963) and Carstens (1966). This approximation also significantly simplifies equation (3) while still retaining the key frazil-ice dynamics. Averaging equation (3) over the mixed layer of depth $D$ yields

$$\frac{\partial n}{\partial t} = -\frac{\partial}{\partial R}(Gn) - \gamma n + \dot{N}\delta(R), \tag{13}$$

where $\gamma = W/D$.

The temperature of the mixed layer or tank evolves due to heat extraction to the atmosphere per unit volume $Q$ and release of the latent heat of solidification

$$\rho_l c_l \frac{dT}{dt} = -Q + 2\pi \text{Nu} k_l \Delta T \int_0^\infty fnR\,dR. \tag{14}$$

There is an implicit assumption that the ice removed through gravitational settling does not inhibit heat loss to the atmosphere (by ice accumulation at the surface), otherwise $Q$ would decrease over time.



## 3.2 Rapid growth – the frazil-ice explosion

In a typical experiment, a relatively small number of crystals are seeded into supercooled water, for example by running a saw blade over a block of ice. Over time, the number of crystals undergoes a period of rapid growth, producing an optically dense suspension (Hanley and Tsang, 1984). Svensson and Omstedt (1994) include a figure from Daly (1992, citing personal communication) showing a period of rapid growth in the number of crystals: the total number of crystals increased by four orders of magnitude over around 250 s. The frazil-ice explosion was observed to reduce the supercooling in the mixed layer to a small residual amount.

Our goal in this section is to ascertain the conditions under which such a frazil-ice explosion can occur, and hence determine conditions for their occurrence in geophysical settings as well as laboratory experiments. To motivate our approach, we consider the time evolution of a mixed layer seeded with some small initial concentration and cooled beneath freezing by a constant flux $Q$. The initial size distribution of crystals is taken to be uniform on $[0, 2R_0]$, and we vary the total number of crystals to vary the initial concentration. Throughout this section, we fix the crystal growth law $f_2 = 1$. We present an example of such a calculation in Figure 2. In one calculation, with slightly less ice initially present (blue curve in Figure 2), all of the ice is removed (by gravitational rise) and supercooling continues to build. Eventually we would expect heterogenous and later homogenous nucleation to occur (Daly, 1984), but we do not model these processes. In the other calculation, with slightly more ice initially present (red curve in Figure 2), the ice concentration increases rapidly before attaining a steady state in which supercooling is almost exhausted (see section 3.4). We consider this an example of a 'frazil-ice explosion' of the kind observed in experiments. A greater initial seeding concentration of ice always makes an explosion more likely, so we investigate the minimum initial concentration (or equivalently number of crystals, if the initial size is fixed) required to trigger an explosion as a function of the other parameters of the system.

We summarize the results of our investigations in Figure 3. Increasing the turbulent intensity $\epsilon$ (Figure 3a) increases the rate of secondary nucleation, since crystals are more likely to collide, which promotes frazil explosions. Increasing the mixed-layer depth $D$ (Figure 3b) reduces the rate at which crystals are removed gravitationally, which again promotes frazil explosions. A slightly weaker effect (note the different scale on the axis) is that increasing the cooling rate $Q$ (Figure 3c) promotes frazil explosions. The direct mechanism is that higher cooling promotes ice growth, increasing the frazil concentration. However, there is also an important indirect mechanism: ice growth shifts the crystal size distribution to larger crystal sizes, which are more likely to collide, leading to greater secondary nucleation. This effect is somewhat offset by the fact that larger crystals are also more effectively removed by gravitational rise.

These mechanisms can be understood more quantitatively by scaling analysis. First, we integrate equation (13) across crystal sizes to obtain an evolution equation for the total number density of crystals (recalling the growth shifts the size distribution but doesn't change the total number of crystals)

$$\frac{dN}{dt} = \dot{N} - \int_0^\infty n\gamma \, dR. \tag{15}$$



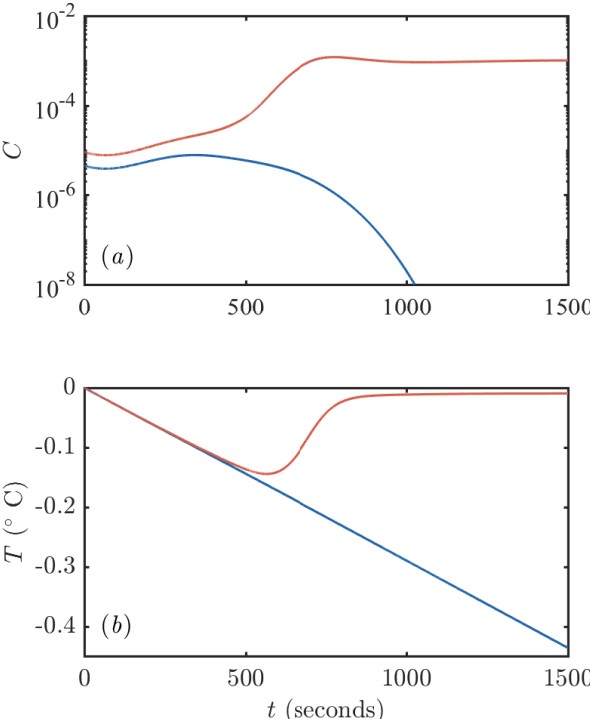

**Figure 2.** Example of the evolution of (a) frazil-ice concentration and (b) supercooling after an initial seeding event. One calculation is initialized with slightly more crystals than the other: the red curve has an initial crystal number density of $10^6 \, \mathrm{m}^{-3}$ compared to $5 \times 10^5 \, \mathrm{m}^{-3}$ for the blue curve. The former leads to a frazil-ice explosion with a large concentration of ice and all the supercooling exhausted. The latter eventually loses all of the ice initially present. Calculations were performed with $Q = 1200 \, \mathrm{W \, m}^{-3}$, $\epsilon = 5 \times 10^{-3} \, \mathrm{m}^2 \, \mathrm{s}^{-3}$, $D = 1 \, \mathrm{m}$, $\tilde{n}_{\max} = 4 \times 10^6 \, \mathrm{m}^{-3}$, $R_0 = 0.2 \, \mathrm{mm}$, $H = 0.05 \, \mathrm{mm}$. The parameter values are similar to Svensson and Omstedt (1994).

If gravitation removal were to act alone, we find that

$$\frac{dN}{dt} = -\frac{W_0}{D}\overline{R}N, \tag{16}$$

where $\overline{R}$ is the mean crystal size. Thus crystals are removed exponentially on a settling timescale $\tau = D/W_0\overline{R} \approx 300 \, \mathrm{s}$ (based on $D = 1$ and $\overline{R} = 0.2 \, \mathrm{mm}$, the initial average crystal radius), which is commensurate with the evolution timescale observed in Figure 2.

Second, we consider a balance between secondary nucleation and gravitational removal. We expect a frazil explosion when the secondary nucleation (equation 8) is much greater than gravitational removal:

$$N^2 U_0 \pi \overline{R}^3 \gg \frac{N W_0 \overline{R}}{D}, \tag{17}$$

$$\Rightarrow N \gg N_{\mathrm{crit.}} \sim \frac{W_0}{U_0 \overline{R}^2 D}. \tag{18}$$

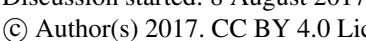



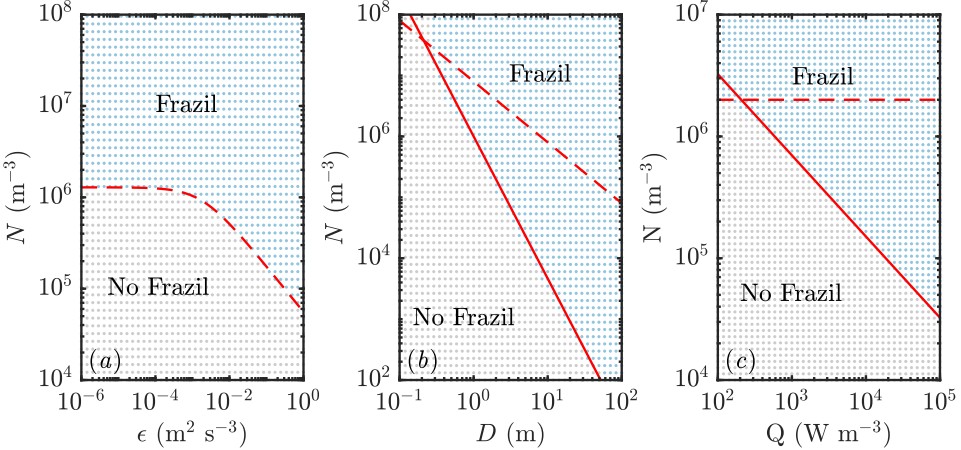

**Figure 3.** Regime diagram showing how the parameters of the system affect the likelihood of a frazil explosion (coloured blue and labelled 'frazil' in each panel) or collapse of the ice population by gravitational settling (coloured grey and labelled 'no frazil'). Each dot represents a separate numerical simulation. Increased (a) turbulent intensity $\epsilon$, (b) mixed-layer depth $D$, and (c) cooling rate $Q$ all promote frazil explosions. Apart from the panels in which they are varied, the parameters used are as in Figure 2. The dashed and solid curves show the predictions of scaling analysis described in the main text. The dashed curves corresponds to equation (18) and the solid curves to equation (22). Note that in panel (a) these equations give the same predictions, so only one curve is plotted.

If $\overline{R}$ is given by the initial average crystal radius, then in terms of the external parameters of the system shown in Figure 3, we would naively expect $N_{\mathrm{crit.}}$ to decrease with turbulent intensity (inversely proportional to $U_0$), mixed-layer depth (inversely proportional to $D$), and be independent of $Q$. The first prediction (Figure 3a) is supported by the numerical results. However, the second prediction (Figure 3b) and third prediction (Figure 3c) are not (the dashed curves do not agree with the numerical

5    results). The resolution of these discrepancies lies in recognising that the average crystal size $\overline{R}$ is not a constant external parameter (i.e. set by the initial condition as a consequence of the seeding strategy), but rather depends on crystal growth.

We now suppose that the average crystal size is determined by the amount a crystal can grow over a crystal removal timescale $\tau$, i.e.

$$\overline{R} \sim G\tau. \tag{19}$$

10    This is a good approximation provided $G\tau$ is much larger than the initial crystal size. The growth rate $G$ is proportional to the supercooling, in particular $G = \mathrm{Nu} f \, k_l \Delta T/(\rho_i L H)$. We can estimate the supercooling from the heat balance equation (14) in which the crystal growth term is negligible until the frazil explosion occurs. We find

$$\rho_l c_l \Delta T \sim Q\tau,$$
$$\Rightarrow G \sim \frac{\mathrm{Nu} f \, k_l Q\tau}{\rho_l c_l \rho_i L H}. \tag{20}$$



We substitute equation (20) into equation (19) and recall that $\tau = D/W_0\overline{R}$. Rearranging for $\overline{R}$ we find

$$\overline{R}^3 \sim \frac{\mathrm{Nu} f k_l Q}{\rho_l c_l \rho_i L H} \left(\frac{D}{W_0}\right)^2.  \tag{21}$$

We then substitute this estimate for $\overline{R}$ into equation (18) and obtain

$$N_{\mathrm{crit.}} \sim \frac{1}{U_0} \left(\frac{W_0}{D}\right)^{7/3} \left(\frac{\mathrm{Nu} f k_l Q}{\rho_l c_l \rho_i L H}\right)^{-2/3}.  \tag{22}$$

Equation (22) is very appealing because it can explain nearly all the results presented in Figures 3a, 3b and 3c (the solid curves agree with the numerical results much better than the dashed curves). The heat flux result is perhaps slightly affected by the initial crystal size distribution at small $Q$, but overall the agreement is very good.

In terms of our crystal growth rate, our scaling argument in equation (22) suggests that the faster growth laws would necessitate a smaller initial concentration of ice to trigger a frazil explosion, something that we observe in numerical experiments
(*cf.* Figure 5).

In conclusion, we find that the explosive growth of frazil ice requires a sufficiently large numbers of seed crystals (perhaps supplied from the atmosphere as sea spray freezes, broken off from pieces of an ice shelf above a plume, or sediment acting as nuclei for crystal growth). Such growth is promoted by high turbulent intensity, a deeper mixed layer, and strong cooling rate (or larger seed crystals).

**3.3  Transient evolution**

Figure 4 shows an example of how the crystal size distribution (CSD) evolves when a frazil explosion occurs. Initially, the larger seed crystals are removed gravitationally, while crystals are nucleated at the smallest size due to collisional breeding. These crystals grow. Note the 'travelling wave' type solutions evident at 100s and 200s with the radius of crystals increasing over time. Indeed, there are travelling wave solutions to equation (13) if crystal growth is the only process that affects the CSD
evolution. Finally, a steady-state distribution is achieved, which we discuss in more detail in section 3.4.

We next consider the impact of different parameterisations of crystal growth $f_{1-3}$. One main experimental measurement is mixed-layer temperature as a function of time. We find that this observable is sensitive to the crystal growth rate, as shown in Figure 5. Faster crystal growth means a faster increase in crystal concentration, with the peak growth rate occurring several hundred seconds earlier. This in turn means that the peak supercooling is lower, because of the latent heat liberated by crystal
growth. These differences are experimentally detectable.

Our new parameterization produces broadly similar transient evolution curves to the older model of Svensson and Omstedt (1994). It is therefore encouraging to note that Svensson and Omstedt (1994) were able to use their model to explain the experimental observations of degree of supercooling. However, demonstration of consistency with experiments does not conclusively show that a parameterization of crystal growth is correct, because other factors also affect the predicted supercooling,
such as the size distribution of the initial seed crystals (which was not controlled by the experimentalists) as shown in Figure 6. Larger seed crystals grow more slowly and achieve greater maximum supercooling, which produces similar predictions to





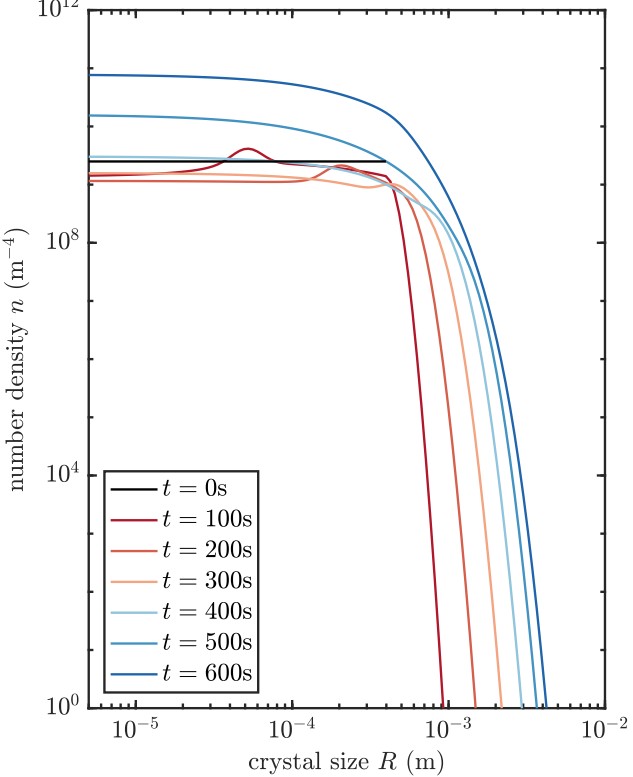

**Figure 4.** Evolution of crystal size distribution for initial conditions that permit a frazil explosion (red curve in Figure 2).

using a slower growth-rate law. This suggests that it is worthwhile for experimentalists to try to measure crystal sizes, as well as supercooling, in order to discriminate between models.

In conclusion, we have shown that crystal growth rate significantly affects the transient evolution of crystal size distribution. Further experimental observations are needed to discriminate between models. Geophysically, we note that the differences between models occur on timescales of a few hundred seconds. This timescale is proportional to mixed-layer depth, so a deeper mixed layer would be associated with even longer transient frazil-ice dynamics. The transient difference are therefore likely to be most significant to systems where the frazil ice is subject to processes that act on similar or shorter timescales to the transient relief of supercooling. (For processes that act on longer timescales, the frazil-ice dynamics would have equilibrated to the steady states discussed in the next section.) For example, a lateral current of $1 \, \mathrm{m/s}$ would take $100 \, \mathrm{s}$ to move material across a lead that is $100 \, \mathrm{m}$ wide. These numbers offer some indication that these transient model differences may well be geophysically significant. Indeed, we show an example in the context of Ice Shelf Water plumes in section 4.



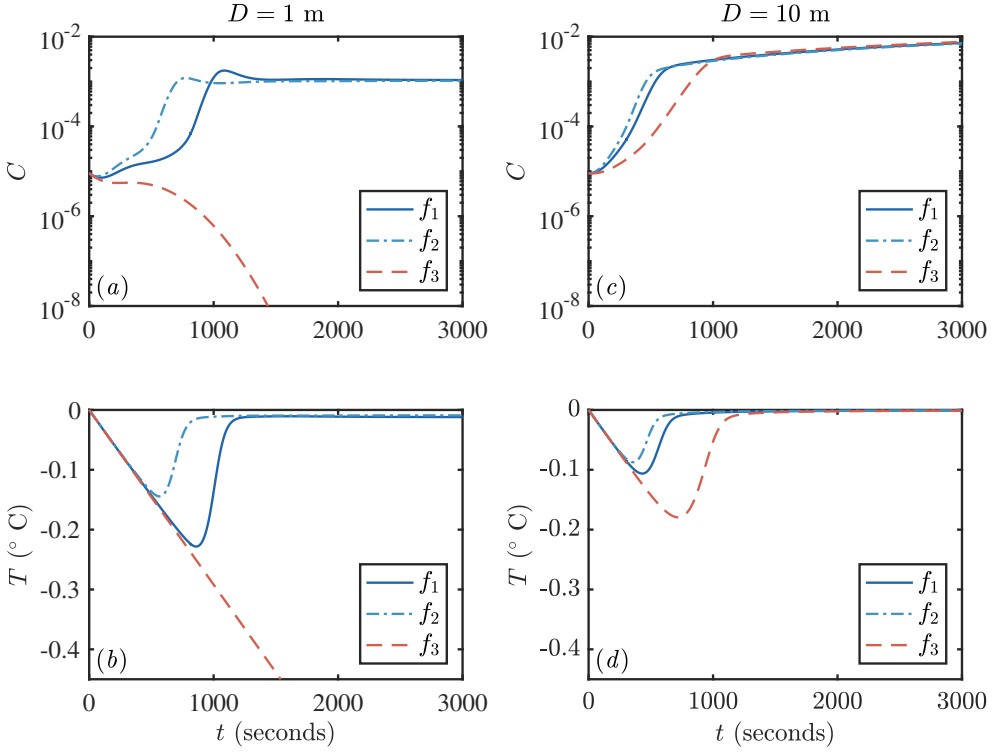

**Figure 5.** Evolution of crystal concentration (a, c), mixed-layer temperature (b, d) using the three growth rate formulae discussed in section 1.2. Using $D = 1$ m leads to frazil explosions for growth laws 1 and 2, but the population collapses for growth law 3. For a deeper mixed layer $D = 10$ m, all the growth laws result in a frazil explosion. Other parameters are as in Figure 2.

## 3.4 Steady states

We observed that the crystal size distribution evolves to a steady state. In this section we study these steady states by numerically integrating our transient model to reach a steady state for each of the three growth laws, and by finding analytical steady-state solutions of the governing equations for two of the growth laws. We present an example of numerically obtained steady states in Figure 7. Changing the growth law subtly shifts the crystal size distribution.

In order to better understand the physical processes involved in maintaining this steady state, we analyse the steady state solutions of equation (13), namely

$$\frac{\partial}{\partial R}(G_0 f n) + \gamma_0 R n = \dot{N} \delta(R) \tag{23}$$

where $\gamma_0 = W_0/D$. We start with the first growth law $f = f_2$ (a constant). First, we integrate equation (23) when $R > 0$ to obtain

$$n = n_0 \exp\left(-\frac{\gamma_0}{2 G_0 f_2} R^2\right). \tag{24}$$





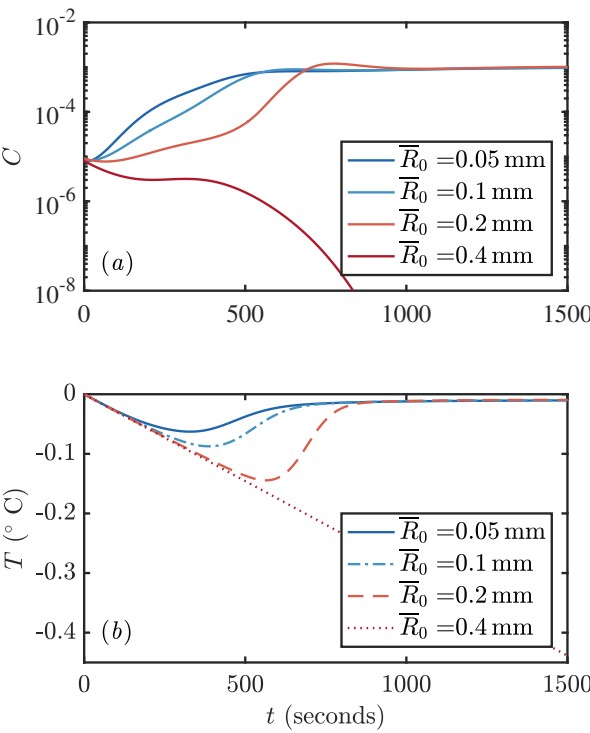

**Figure 6.** Evolution of (a) crystal concentration and (b) mixed-layer temperature using four different initial average crystal sizes (0.05, 0.1, 0.2 and 0.4 mm), growth law $f_2$ and other parameters as in Figure 2.

Second, at $R = 0^+$, equation (7) implies that

$$G_0 f_2 n(R = 0^+) = \pi U_0 \tilde{n}_{\max} \int_0^\infty n(R) R^3 \, dR, \tag{25}$$

where we assume that the total number of crystals exceeds $\tilde{n}_{\max}$ so $\tilde{n} = \tilde{n}_{\max}$. This is reasonable because there is a very large number of crystals after a frazil-ice explosion has occurred. Equation (25) can be manipulated to show that $G_0 f_2 = \gamma_0^2 / 2\pi U_0 \tilde{n}_{\max}$. This expression allows the steady state supercooling to be calculated since $G_0 = \mathrm{Nu} k_l \Delta T / \rho_i L H$. Third, we use the overall heat balance from equation (14) in steady state

$$Q = 2\pi \mathrm{Nu} k_l \Delta T \int_0^\infty f n R \, dR \tag{26}$$

to determine the unknown prefactor $n_0$. Finally, we calculate the average crystal size (mean) $\overline{R}$, the total number of crystals $N$, the total crystal concentration $C$.

We then repeat the analysis for the growth law $f = f_3 \equiv H/R$. We report the results in Table 1.





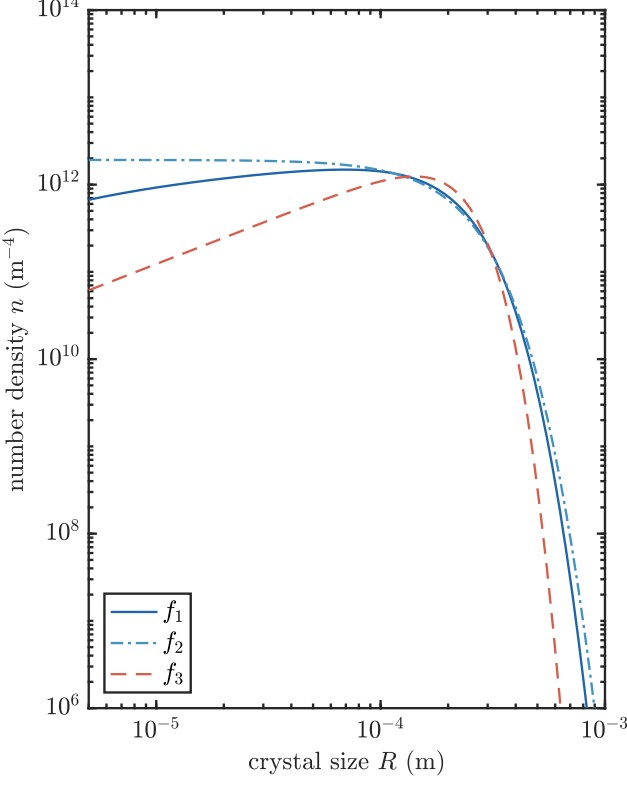

**Figure 7.** Numerically calculated steady-state crystal size distributions using the three growth rate formulae discussed in section 1.2. The parameters are as in Figure 2.

We conclude from this analysis that the average crystal radius is insensitive to the crystal growth rate. This initially surprising result can be understood by considering that the balance between growth and precipitation at large crystal sizes gives $G_0 f \sim \gamma_0 \overline{R}^2$, while the balance between growth and nucleation of the smallest crystals gives $G_0 f \sim U_0 \tilde{n}_{\max} \overline{R}^4$. The growth rate dependent term $G_0 f$ can be eliminated between these equations and

$$\overline{R} \sim \left( \frac{\gamma_0}{U_0 \tilde{n}_{\max}} \right)^{1/2}, \tag{27}$$

in agreement with the expressions in Table 1. Therefore average crystal size depends on (1) turbulent intensity and efficiency of secondary nucleation (more secondary nucleation means smaller crystals), and (2) on gravitational removal (a larger gravitational removal rate prefactor $\gamma_0$ means larger crystals). The first effect is readily understood: secondary nucleation creates tiny crystals. The second is more subtle because gravitational removal tends to remove larger crystals. However, secondary nucleation increases more rapidly as a function of crystal radius than gravitational removal. Thus enhanced gravitational settling enhances the removal of large crystals, and mutes their efficiency in driving secondary nucleation, leading to the scaling given in equation (27). In geophysical settings, the crystal rise velocity, mixed-layer depth and turbulent intensity can be measured much more easily than the efficiency of secondary nucleation. We therefore suggest choosing this parameter to match with





**Table 1.** Steady state crystal size distribution results for two growth laws. . Note that $\Gamma()$ here denotes the Gamma function. The steady state supercooling can be computed using $\Delta T = G_0 \rho_i L H / (\mathrm{Nu} k_l)$

| Quantity | $f = f_2$ | $f = f_3 \equiv H/R$ |
|---|---|---|
| $n$ | $n_0 \exp\left(-\dfrac{\gamma_0}{2 G_0 f_2} R^2\right)$ | $n_0 R \exp\left(-\dfrac{\gamma_0}{3 G_0 H} R^3\right)$ |
| $G_0$ | $\dfrac{\gamma_0^2}{2\pi U_0 \tilde{n}_{\max} f_2}$ | $\dfrac{\gamma_0^{5/2}}{3H\left(\pi U_0 \tilde{n}_{\max} \Gamma(5/3)\right)^{3/2}}$ |
| $n_0$ | $2\pi \dfrac{Q}{\rho_i L H \gamma_0}\left(\dfrac{\gamma_0}{U_0 \tilde{n}_{\max}}\right)^{-2}$ | $\dfrac{9\left(\pi\Gamma(5/3)\right)^{5/2}}{2\pi\Gamma(2/3)}\dfrac{Q}{\rho_i L H \gamma_0}\left(\dfrac{\gamma_0}{U_0 \tilde{n}_{\max}}\right)^{-5/2}$ |
| $N$ | $\pi \dfrac{Q}{\rho_i L H \gamma_0}\left(\dfrac{\gamma_0}{U_0 \tilde{n}_{\max}}\right)^{-3/2}$ | $\dfrac{3\left(\pi\Gamma(5/3)\right)^{3/2}}{2\pi}\dfrac{Q}{\rho_i L H \gamma_0}\left(\dfrac{\gamma_0}{U_0 \tilde{n}_{\max}}\right)^{-3/2}$ |
| $\overline{R}$ | $\dfrac{1}{\pi}\left(\dfrac{\gamma_0}{U_0 \tilde{n}_{\max}}\right)^{1/2}$ | $\dfrac{1}{\Gamma(2/3)\left(\pi\Gamma(5/3)\right)^{1/2}}\left(\dfrac{\gamma_0}{U_0 \tilde{n}_{\max}}\right)^{1/2}$ |
| $C$ | $\dfrac{\pi}{2}\dfrac{Q}{\rho_i L \gamma_0}\left(\dfrac{\gamma_0}{U_0 \tilde{n}_{\max}}\right)^{-1/2}$ | $\dfrac{3\Gamma(4/3)\left(\pi\Gamma(5/3)\right)^{1/2}}{2\Gamma(2/3)}\dfrac{Q}{\rho_i L \gamma_0}\left(\dfrac{\gamma_0}{U_0 \tilde{n}_{\max}}\right)^{-1/2}$ |

observations of average crystal size. For example, choosing the reduced value $\tilde{n}_{\max} = 4 \times 10^5 \ \mathrm{m}^{-3}$ would give an average crystal size of about $0.5 \ \mathrm{mm}$.

The total crystal concentration $C$ is also insensitive to the crystal growth rate. We can show this by continuing our scaling analysis as follows. From equation (26), we estimate

$$Q \sim \mathrm{Nu} k_l \Delta T f \overline{R} N,$$

$$\sim \rho_i L H G_0 f \overline{R} N,$$

$$\sim \rho_i L H \gamma_0 \overline{R}^3 N,$$

$$\sim \rho_i L \gamma_0 \overline{R} C, \tag{28}$$

where we have used $G_0 f \sim \gamma_0 \overline{R}^2$ from the growth versus settling balance. If we define a surface heat flux scale $Q_{\mathrm{surf.}} = QD$, and recall $\gamma_0 \overline{R} = W_0 \overline{R}/D$, we find

$$C \sim \frac{Q_{\mathrm{surf.}}}{\rho_i L W_0 \overline{R}}. \tag{29}$$

Thus at steady state, the total amount of frazil ice is determined by a balance between the surface heat flux and the rate of export of latent heat in the form of frazil ice that is removed gravitationally. This steady state balance is unaffected by crystal growth rate (at least in the absence of advective processes).



## 4 Frazil-laden plume underneath an ice shelf

Frazil ice also forms in plumes of ice shelf water (ISW) beneath floating ice shelves. A full examination of the dynamics of these plumes is beyond the scope of this paper, and we refer the reader to previous studies by Jenkins (1991); Jenkins and Bombosch (1995); Smedsrud and Jenkins (2004). Instead we focus more narrowly by considering a simple case study that

illustrates the possible impact of different treatments of frazil-ice processes on the dynamics of an ISW plume. A linear ice shelf rises from a depth of 1400 m below sea level to a depth of 285 m below sea level over a horizontal distance of 600 km. The ambient seawater is treated as an approximation to High Salinity Shelf Water (HSSW) with a linear stratification. Jenkins and Bombosch (1995) conceived this setting as a simple configuration that is representative of a large Antarctic ice shelf. Frazil ice is nucleated when the plume falls beneath the pressure-dependent freezing temperature. Frazil ice increases the plume

buoyancy and so accelerates the plume. Thus we might naively expect that faster crystal growth would lead to higher frazil concentrations and faster flowing plumes. In this section, we show that this expectation is confounded by complex feedbacks between plume dynamics and frazil-ice processes.

The plume model accounts for the evolution of plume depth $D$, and the depth-averaged plume velocity $U$, temperature $T$ and salinity $S$ as a function of distance $s$ along the ice shelf. The frazil-ice dynamics part of the model is essentially the same

as that described in Eq. 3, but integrated over the depth of the plume. The depth-averaged frazil crystal size distribution evolves according to

$$\frac{\partial(DUn)}{\partial s} = -D\frac{\partial}{\partial R}(Gn) - p(R)n + D\dot{N}\delta(R), \tag{30}$$

where $p(R)$ is the rate at which frazil precipitates onto the base of the ice shelf.

We retain the approach of Smedsrud and Jenkins (2004) as far as possible. The full set of governing equations is described

in that paper. Software code to reproduce the calculations in the paper is available (Rees Jones, 2017). Note that our thermal calculation includes an estimate of the conductive heat flux into the ice shelf (based on Holland and Jenkins (1999), using a core ice-shelf temperature of $-15\,°\mathrm{C}$, A. Jenkins, personal communication). We use a large number of crystal size classes in the discrete calculation (1000), to ensure the crystal size distribution is well resolved. By contrast Smedsrud and Jenkins (2004) use only 10 classes. This affects the quantitative results but not the qualitative behaviour of the system. One important difference

compared to the mixed-layer models (Section 3) is that the precipitation rate depends both on crystal rise velocity and also on the plume velocity, because precipitation from a turbulent plume occurs when crystal buoyancy exceeds the turbulent shear stress acting to keep it in suspension. Thus precipitation occurs when the plume velocity $U$ is less than some critical velocity $U_c$ that can be expressed in terms of a critical Shields number.

### 4.1 Results

The dynamics of an ISW plume can be very sensitive to frazil-ice processes. Our numerical investigation found two basic types of behaviour. Sometimes frazil ice precipitates out over a relatively short distance $O(10\ \mathrm{km})$ and the plume itself is barely affected by the frazil. At other times the frazil ice is sustained over $O(100\ \mathrm{km})$ and plume is rendered more buoyant. We





illustrate this range of behaviour and explain the underlying physical mechanisms by varying the rates of secondary nucleation and crystal growth (Figure 8). In terms of secondary nucleation, we consider no nucleation $\tilde{n}_{\max} = 0 \text{ m}^{-3}$, intermediate nucleation $\tilde{n}_{\max} = 500 \text{ m}^{-3}$ comparable to Smedsrud and Jenkins (2004), and high nucleation $\tilde{n}_{\max} = 4 \times 10^6 \text{ m}^{-3}$ comparable to Svensson and Omstedt (1994). In terms of crystal growth, we contrast a slow growth law (Smedsrud and Jenkins, 2004) and a

fast growth law (Svensson and Omstedt, 1994), and use the crystal geometries assumed in these papers. Calculations with the logarithmic growth law $f_1$ introduced in Section 1.2 are very similar to the case of fast crystal growth.

Our sensitivity experiments (Figure 8) show that secondary nucleation is needed to sustain the frazil ice population. We would also expect a continuous source of small seed crystals to have a similar effect, were the source sufficiently large. In calculations without nucleation, the crystals precipitate out and the total concentration remains small, insufficient to affect

the plume dynamics. The faster growing crystals precipitate more over a shorter distance (dashed blue curve, panel $e$), because larger crystals rise faster and are more difficult to keep in suspension. After the frazil ice precipitates out of the plume, supercooling increases (blue curves, panel $d$), leading to a high rate of direct basal freezing (blue curves, panel $g$).

By contrast, a high nucleation rate triggers rapid growth of frazil ice, which relieves the supercooling in the plume (red curves, panels $c$, $d$, $f$). This behaviour is analogous to the 'frazil-ice explosion' we observed previously (Section 3.2), and

occurs when secondary nucleation exceeds crystal removal by precipitation. The increased frazil concentration leads to a more buoyant plume, causing it to accelerate and narrow slightly (red curves, panels $a$, $b$). Precipitation is relatively unimportant (red curves, panel $e$) as a result of a positive feedback: a faster flowing plume keeps crystals suspended more easily. Furthermore, nucleation produces small crystals, which again are kept in suspension more easily. A faster crystal growth rate is associated with a faster increase in crystal concentration along the slope, although the quasi-steady state achieved after the supercooling

is almost exhausted is similar. As we found previously (equation 29), the quasi-steady ice concentration reflects the overall energy balance of the system, rather than the growth dynamics.

The case of intermediate nucleation rate illustrates the surprising interplay between nucleation, growth and precipitation of crystals. The calculation with a faster growth rate initially leads to a greater concentration of frazil ice, but the ice concentration is eventually overtaken by the slower growth rate calculation (green curves, panel $c$). Faster growth leads to larger crystals

which in turn are more readily precipitated (dashed green curve, panel $e$). This means that the crystal concentration eventually decreases, reducing the plume buoyancy and causing it to decelerate (dashed green curve, panel $a$). In this case, the plume thickness starts to increase rapidly as the plume begins to intrude at depth (dashed green curve, panel $b$). By contrast, the case of slower growth rate eventually reaches a crystal concentration comparable to the calculations with larger crystal nucleation rate.

In terms of the large-scale dynamics, different parameterizations of crystal growth rate and nucleation can be the difference between a plume that is reinvigorated by frazil ice and reaches the end of the shelf and a plume that decelerates and intrudes at depth. This behaviour is likely to affect the ocean circulation and water mass transformation in the shelf seas around Antarctica. The differences between models could in principle be observed by considering the amount of frazil precipitation relative to basal freezing. The total amount of frazil formation also differs between the models (panel $g$). These differences are surprising:

faster growth can lead to less total frazil-ice formation in total, if it is removed from suspension before it can multiply. This



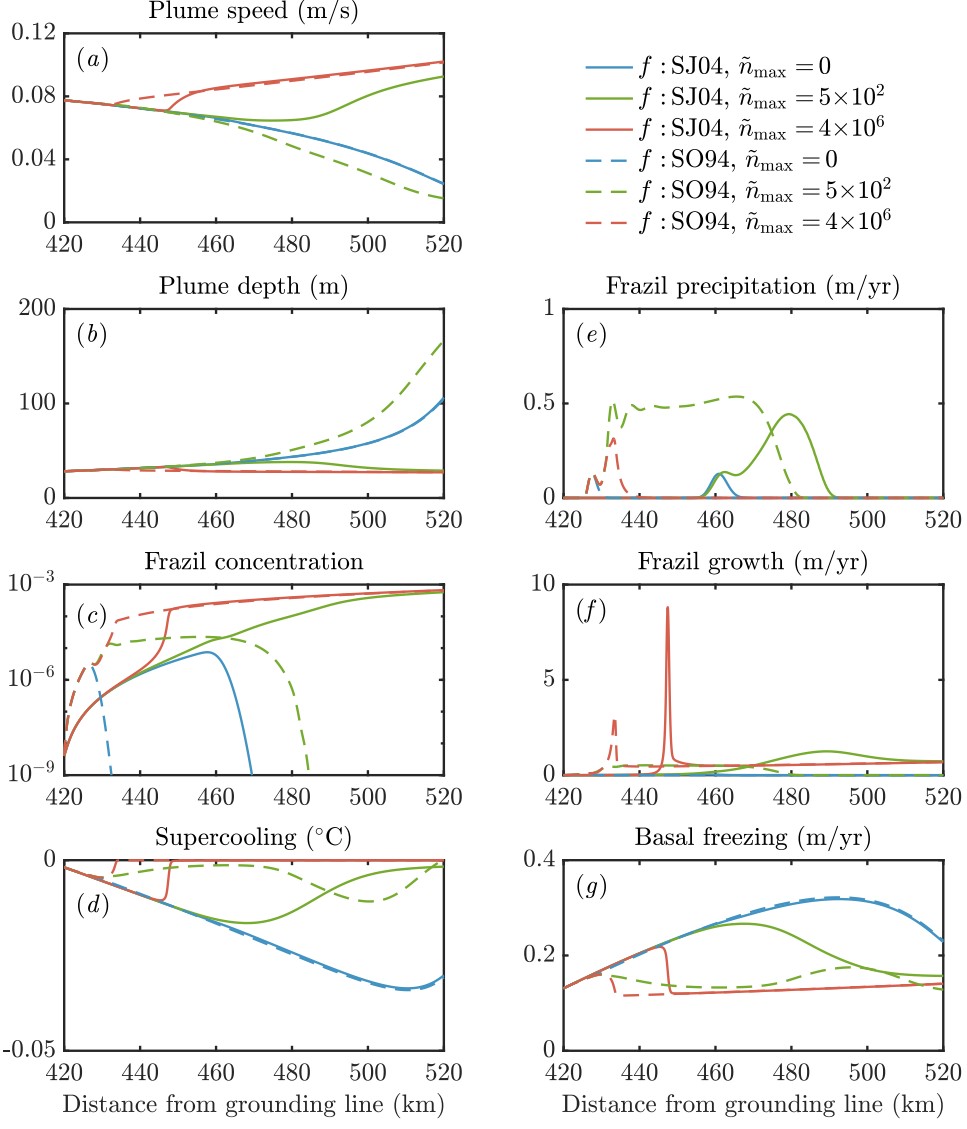

**Figure 8.** The sensitivity of the dynamics of a frazil-laden plume to parameterizations of crystal growth and nucleation. We perform calculations with no secondary nucleation (blue), intermediate nucleation (green) and high nucleation (red). Solid lines denote slow crystal growth SJ04 (Smedsrud and Jenkins, 2004) and dashed lines denote fast crystal growth SO94 (Svensson and Omstedt, 1994). Calculations with the logarithmic growth law $f_1$, introduced in Section 1.2 based on Rees Jones and Wells (2015), are very similar to the SO94 results. Note that the solid red curve in panel ($e$) is approximately zero.



suggests that small-scale frazil-ice processes, which are hard to constrain in models, can have major implications for our understanding of the dynamics of plumes of ISW beneath Antarctica's floating ice shelves.

## 5   Conclusions

The theory of frazil-ice dynamics pioneered by Daly (1984) encompasses the nucleation, growth and removal of frazil ice. It describes the evolution of the size distribution of a population of crystals. We have applied this theory to understand ice formation in a supercooled ocean mixed layer and in a plume of ISW underneath a floating ice shelf. Understanding frazil-ice processes is significant to our understanding of ice–ocean interaction in the earliest, most explosive phase of ice growth. We have identified critical conditions for a self-sustained frazil-ice explosion, which occurs when secondary nucleation exceeds crystal removal. Crystal growth rate affects such explosions by changing the crystal size distribution, and also alters the transient evolution of frazil ice, promoting faster increases in frazil concentration. We determined steady-state crystal size distributions, and found that these were relatively insensitive to crystal growth rate, but sensitive to secondary nucleation and crystal removal. Thus measurement of crystal sizes could be used to estimate the nucleation rate indirectly. Finally, we showed that the parameterization of crystal growth rate and nucleation can dramatically affect the fate of plumes of supercooled ice shelf water, with implications for ice accretion on ice shelves and ocean circulation. Although our understanding of crystal growth rate has advanced recently, our understanding of crystal nucleation remains limited. Our calculations suggest that this is potentially a significant uncertainty, and is a topic ripe for future research.

*Code availability.*   Please see https://github.com/davidreesjones/frazil-dynamics for software code to reproduce calculations and figures in the paper (Rees Jones, 2017).

*Competing interests.*   We have no competing interests to declare.

*Acknowledgements.*   We thank M.G. Worster for comments on an earlier version of this manuscript, and A. Jenkins for discussing previous models of frazil-laden plumes. This publication arises from research funded by the John Fell Oxford University Press (OUP) Research Fund, and A.J.W. also acknowledges financial support through the research program of the European Union FP7 award PCIG13-GA-2013-618610 SEA-ICE-CFD. D.R.J. acknowledges research funding through the NERC Consortium grant NE/M000427/1 and NERC Standard grant NE/I026995/1. We would like to thank the Isaac Newton Institute for Mathematical Sciences for its hospitality during the programme Melt in the Mantle which was supported by EPSRC Grant Number EP/K032208/1.



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
