# Peer review of "Frazil-ice growth rate and dynamics in mixed layers and sub-ice-shelf plumes"

_The Cryosphere, 2017_

## Referee Comment (RC1) · Anonymous Referee #1 · 6 Sep 2017

This paper presents theoretical ideas that are topical and the subject of a variety of recent modeling papers of interest to The Cryosphere. The writing is extremely clear, concise and the free from errors. This clarity meant that I learned a good deal from this paper, and its abstract and conclusions are particularly succinct. The authors review the topic fairly, assigning proper credit.

The authors introduce three different parameterizations of crystal growth, and nicely fit previous work (including their own) into these three parameterizations. They then compare the effect of these parameterizations on the dynamics of overall ice production. Numerical methods are used to solve the governing equations, and the new approach

essentially increases the number of crystal size classes of a previous model to pseudo-continuous. They then consider various aspects of the growth of a body of frazil ice in a "well mixed" layer and in a buoyant plume.

The only two substantive comments I have relate to the later sections of the paper. First, in section 3, I would have liked a very precise definition of what the authors mean by a "mixed layer". I assume temperature and (if appropriate) salinity are uniform in their mixed layer. But what about velocity, or is a stagnant layer only being considered? The authors mention that they take depth-averaged frazil concentration in the plume model, but what is the assumption for their well mixed layer?

Second, in abstract the authors state that they "apply our model. . .. to a buoyant plume under a floating ice shelf. " However I was confused about what aspect of their model they were applying since Fig 8 seems to be a sensitivity test to changes in parameters in SJ04 and SO94. It is not clear to me how the development in sections 1-3 are incorporated into the buoyant plume model, and I was disappointed that this was not demonstrated more explicitly. Perhaps this could be easily clarified by explaining how f1 to f3 fit within SJ04 and SO94.

Technical Corrections p. 3, line 25: Does the expression for Nu come from Galton-Fenzi et al (2014)?

p. 3, Line 26: "diffusivity of salt in water"

p. 4, line 9: I would have found it reassuring to be told that C is the volume occupied by ice crystals per unit volume of mixture

p. 5, line 29: please comment on the consequences of ignoring flocculation and crystal break-up.

p. 5, line 32: possibility of confusion due to use of $\delta$ for delta function and thermal boundary layer, possibly resolved by a subscript on thermal boundary layer.

p. 6, line 1: possibility of confusion due to use of $\varepsilon$ for turbulence intensity.

p. 6, line 15" "calibration"

p. 7: section 3: Please define precisely what you mean by a "mixed layer" in terms of profiles of temperature, salinity, velocity, ice crystal concentration, etc. How would the presence of waves affect your "mixed layer"?

p. 9, line 4: "D = 1 m" I assume it is 1 m?

p. 11, line 13: Contradictory statement that sediment would act as nuclei since you state on p.6, line 5 that Daly expects this to be unlikely.

p. 11, line 26-31: "experimentalists"? Which experiments are you referring to in this paragraph?

p. 12, line 9: 1 m/s is a rather rapid current for the ocean.

Fig 5 & Fig 6: cropping of symbols on abscissa

p. 13, line 9: why is f2 called the first growth law? Is it not true that f2 $\approx$ 1, rather than « 1 (see p.3)

Fig 6 caption: by "crystal size" I believe you mean crystal radii?

p. 14, line 4-5: Please outline how equation (25) is manipulated. I assume equation (24) is substituted into (25) and integrated?

p. 15, line 6: please remind the reader that U} indicates turbulence intensity and nmax indicates secondary nucleation.

p. 15, line 7-13: is it possible to discuss the experimental behavior of secondary nucleation in relation to the model predictions?

p. 16, Table 1: it would be helpful to have the parameters names on the table as a reminder to the reader.

p. 17, Section 4: As stated above I did not understand because it seemed to me that the authors had introduced three growth models, and then did not use any of them in

their frazil-laden plume study. Instead they ran sensitivity tests on two "old" models.

p. 18, line 16: "narrow slightly"? Do you mean D becomes slightly smaller?

p. 18, line 20: supercooling is "similar" to what?

p. 18, line 34: "(panel g)"? Do you mean panel c?

p. 19, Fig 8: why is f in the legend since it does not seem to change between model runs? Should it say f1 or f2 or f3?

p. 19, Fig 8: In the caption we are told that calculations with f1 are similar to SO94. But surely it would be more relevant to this paper to show f1 results, rather than a sensitivity study of an "old" model?

p. 18 & 19, Fig 8: I suggest reminding the reader which is the "slow" growth and which is the "fast" growth model by explicitly stating this in the legend. I got rather confused in the discussion on p. 18.
* * *

---

## Referee Comment (RC2) · Anonymous Referee #2 · 3 Oct 2017

This is an interesting and well-written paper that discusses the modelling of frazil ice formation in mixed layers at the surface of the ocean and beneath ice shelves. The presentation is generally clear and logically structured, and should be easy to follow for those familiar with earlier literature. If I have one slight criticism, it is that there are a few key papers that form the background to this work which the authors assume the reader will already be familiar with or will read alongside their paper. To a certain extent that is inevitable, and should not really be a problem, but I personally cannot easily access at least one of the key papers that I do not know, and I suspect that many might be in a similar position. While I am not suggesting adding greatly to the size of this paper with lengthy reviews of earlier work, there are a couple of places where I think a few

extra details would help. Other than that I found little to fault and would suggest that the paper is acceptable with minor revision along the following lines:

1) I found the introduction to the crystal growth parameterisations on page 3 (lines 1-20) unnecessarily opaque. I think a few more details of the Rees Jones and Wells (2015) work may have helped. I assume that the discussion of the heat flux around a disc-shaped crystal, and the relevant boundary layer thickness, at the top of page 3, comes from that work, as does the expression for f1 that comes a little later. I think a few details (with maybe a diagram) of the temperature distribution around a growing disc, to give some insight into where f1 comes from would be really informative. Also it would seem more logical to discuss this first, before the approximations, especially if it is to be denoted by the number 1. (Later on page 13, line 9, f2 is somewhat confusingly referred to as the "first" growth law, presumably because of this slightly illogical sequencing that sees f1 introduced last.) With the temperature distribution around a growing disc described and the correct boundary layer scaling justified, the approximations f2 and f3 can then be put in a better context. Don't they come from the assumption of a spherically symmetric temperature distribution, around the disc edge in the case of f2 and around the entire disc in the case of f3? 2) Sections 3 and 4 discuss applications to two geophysical situations, where an assumption has been implicitly made that all properties are either well-mixed or follow some simple self-similar shape, so that depth-integration produces simple depth-averaged properties (and products of properties). While such an assumption is quite common, it is more questionable in this case than is usual. The term $\Delta T$ in equation (14) has a pre-defined depth-dependence. Even if the layer is uniform in temperature, the super-cooling will be a linear function of depth within the layer, because the freezing temperature is pressure-dependent. Furthermore, the ice concentration cannot be well-mixed, because the distribution of crystals will be determined not just by turbulent diffusion, but also by their buoyant rise. So the concentration will be highest at the top of the layer where the super-cooling will also be a maximum. The use of depth-mean quantities is common in the literature, but I think it would be worthwhile to point out the limitations of the assumption and

the potential impact on the results. For example, could the finding that increasing D promotes "frazil explosions" (page 8, lines 22-23 and figure 3b) be an artefact of this assumption?

3) In section 4 a few words about the plume model behaviour might help to put the results presented in figure 8 into context. Referring the reader to earlier publications for much of the detail is fine, but introducing the basic concept of a buoyant flow generated by melting that subsequently becomes supercooled because of the fall in pressure as it ascends the ice shelf base would be helpful. Also it would be worthwhile pointing out that, since the plume flows along an ice-ocean boundary, it drives direct freezing onto the boundary as well as growth of frazil crystals that can be deposited if the plume flow is weak enough. This would not require many extra words but would clarify for the reader new to the concepts what the three panels on the right-hand side of figure 8 actually show.

4) Finally, some more minor comments:

Page 1, line 17: ". . . phase of ice growth in turbulent waters."

Page 1, line 20: ". . . occurs when it is cooled efficiently . . ."

Page 2, line 6-7: Actually Engelhardt and Determann (1987) did not drill a borehole through the ice, nor did they observe the granular texture of the ice. They used a hot-water drill, so could not recover any samples, although did infer that the bottom 35 m of the ice shelf consisted of a slushy layer of unconsolidated frazil ice. The ice at the bottom of Ronne Ice Shelf was not sampled until a little later (Oerter et al., 1992, Nature, 358, 399-401).

Page 2, line 31: ". . . crystal-mass growth, with the ice crystal . . ."

Page 12, line 6: ". . . transient differences are therefore . . ."

Page 13, line 6: " . . . order to understand better the physical . . ."

Page 17, line 9: See earlier comments (2). You mean when the plume becomes supercooled in a depth-averaged sense?

Page 17, line 20-23: Because you only discuss freezing, this comment seems a little out of place. Don't Holland and Jenkins (1999) set the conductive flux to zero when there is freezing at the ice shelf base? Nevertheless, this comment might fit into a slightly expanded description of the plume (see earlier comments (3)).

Page 17, line 32: "... over O(100 km) and the plume is ..."

Page 18, line 34: In the parentheses, should that be "(panel f)"?

Page 18, first paragraph and caption to figure 8: By "fast" and "slow" growth, do you mean f2 and f3? Perhaps you could clarify the point.

Page 18, line 35: "... can lead to less frazil-ice formation in total, ..."

---

## Author Comment (AC1) · 6 Nov 2017

Please see the attached zip file which contains:

1) A pdf file (ResponseToReferees-TC-2017-155.pdf) with the referees' comments, the authors' detailed responses, and extracts detailing changes made in the revised manuscript.

2) A pdf file (Tracked-Changes-TC-2017-155.pdf), showing changes between the revised manuscript and the original submitted manuscript, including an additional figure (1), and a revised figure (9, previously numbered figure 8).

[Figure]

Please also note the supplement to this comment:
https://www.the-cryosphere-discuss.net/tc-2017-155/tc-2017-155-AC1-supplement.zip

---

## Author Response (AR1)

**Response to Referees:**
**Frazil-ice growth rate and dynamics in mixed layers and sub-ice-shelf plumes**

David W. Rees Jones[1,2] and Andrew J. Wells[1]

[1]Atmospheric, Oceanic and Planetary Physics, Department of Physics, University of Oxford, Clarendon Laboratory, Parks Road, Oxford, OX1 3PU, UK.
[2]Department of Earth Sciences, University of Oxford, South Parks Road, Oxford, OX1 3AN, UK.

*Correspondence to:* David W. Rees Jones (David.ReesJones@earth.ox.ac.uk)

**Notes:** Blue denotes the referee's comment; black denotes our response; red denotes an extract from the revised manuscript. Please see the 'tracked changes' version of the manuscript for a comprehensive list of changes. Note that figure numbers have changed in revision.

**1 Referee 1 (R1)**

5   **R1:** This paper presents theoretical ideas that are topical and the subject of a variety of recent modeling papers of interest to The Cryosphere. The writing is extremely clear, concise and the free from errors. This clarity meant that I learned a good deal from this paper, and its abstract and conclusions are particularly succinct. The authors review the topic fairly, assigning proper credit.

    The authors introduce three different parameterizations of crystal growth, and nicely fit previous work (including their own)
10  into these three parameterizations. They then compare the effect of these parameterizations on the dynamics of overall ice production. Numerical methods are used to solve the governing equations, and the new approach essentially increases the number of crystal size classes of a previous model to pseudocontinuous. They then consider various aspects of the growth of a body of frazil ice in a "well mixed" layer and in a buoyant plume.

    We thank the referee for their very positive assessment of our paper, and for their helpful comments, which have helped us
15  to improve our manuscript further. We address all comments and suggestions below.

  **R1:** The only two substantive comments I have relate to the later sections of the paper. First, in section 3, I would have liked a very precise definition of what the authors mean by a "mixed layer". I assume temperature and (if appropriate) salinity are uniform in their mixed layer. But what about velocity, or is a stagnant layer only being considered? The authors mention that they take depth-averaged frazil concentration in the plume model, but what is the assumption for their well mixed layer?

20   Thank you for this comment; we added a precise definition of what we mean by a "mixed layer". We do not consider the layer to be stagnant, but rather stirred by turbulent fluid flow (for example, wind-driven mixing in the ocean or mechanical stirring in the laboratory experiments). At the start of section 3.1, we added:

    We assume that background turbulent stirring is sufficient to keep the layer well-mixed such that all physical quantities (temperature and crystal size distribution) are uniform over the layer. Such turbulence might be driven,
25  for example in the oceans, by wind, waves, and buoyancy-driven convection. A turbulent flow is mechanically driven in laboratory experiments. Thus we only need to solve evolution equations for average physical quantities across the layer.

  **R1:** Second, in abstract the authors state that they "apply our model. . ... to a buoyant plume under a floating ice shelf." However I was confused about what aspect of their model they were applying since Fig 8 seems to be a sensitivity test to

changes in parameters in SJ04 and SO94. It is not clear to me how the development in sections 1-3 are incorporated into the buoyant plume model, and I was disappointed that this was not demonstrated more explicitly. Perhaps this could be easily clarified by explaining how f1 to f3 fit within SJ04 and SO94.

Thank you for this excellent point. We decided to change this figure (previously numbered Figure 8, now Figure 9) by replacing the SO94 calculations by calculations with the Rees Jones and Wells 2015 (RJW15) growth law (called $f_1$). SJ04 is one of a class of growth laws we label $f_3$, and SO94 is an example of an $f_2$ type growth law (which gives similar results to $f_1$). Note that in the model of SJ04, a constant aspect ratio of 0.02 is assumed, whereas in SO94 and RJW15, a constant thickness 0.05 mm is assumed. In light of these changes to the main text and figures, we didn't change the abstract. However, in the first paragraph of section 4.1, our revised text now reads:

In terms of crystal growth, we contrast a slow growth law and a fast growth law. For a slow growth law, we use **?**, one of the class of growth laws we labelled $f_3$ previously. For a fast growth law , we use **?**, labelled $f_1$ previously. Calculations with the growth law $f_2$ introduced in Section **??** are very similar to the results with $f_1$.

The revised caption of figure 9 (renumbered version of figure 8) now reads:

The sensitivity of the dynamics of a frazil-laden plume to parameterizations of crystal growth and nucleation. We perform calculations with no secondary nucleation (blue), intermediate nucleation (green) and high nucleation (red). Solid lines denote slow crystal growth SJ04 (Smedsrud and Jenkins, 2004), one of the class of growth laws we labelled $f_3$ previously. Dashed lines denote fast crystal growth RJW15 (Rees Jones and Wells, 2015), previously labelled $f_1$. Calculations with the growth law $f_2$ based on SO94 (Svensson and Omstedt, 1994) are very similar to the RJW15 results. Note that in the model of SJ04, a constant aspect ratio of 0.02 is assumed, whereas in SO94 and RJW15, a constant thickness 0.05 mm is assumed. Note also that the solid red curve in panel (*e*) is approximately zero.

**R1:** Technical Corrections p. 3, line 25: Does the expression for Nu come from Galton-Fenzi et al (2014)?

Not exactly. The expressions are very similar because $1/(1+x) \approx 1-x$ when $x$ is small. The prefactor $1.4$ in our expression comes from numerical calculations, whereas Galton-Fenzi use a scaling argument. We tried to clarify our writing:

A good approximation based on our numerical calculations is Nu =...

**R1:** p. 3, Line 26: "diffusivity of salt in water"

Changed as suggested.

$D_S$ is the diffusivity of salt in water

**R1:** p. 4, line 9: I would have found it reassuring to be told that C is the volume occupied by ice crystals per unit volume of mixture

Thanks for this suggestion, we added the following sentence:

> Note that $C$ is the volume occupied by ice crystals per unit volume of mixture.

**R1:** p. 5, line 29: please comment on the consequences of ignoring flocculation and crystal break-up.

There are considerable uncertainties in understanding these processes, as we discuss in the paper (section 2.2). We added a comment about the possible consequences of neglecting flocculation and crystal break-up with the segment now reading:

> In view of the considerable uncertainties in parameterizing flocculation, we neglect this process in all of our calculations ($B = 0$). Indeed, even the sign of $B$ is uncertain, as it not clear whether flocculation or break-up dominates (and the balance of these processes may well depend on the fluid dynamical conditions). If break-up dominates (perhaps in more turbulent environments), setting $B = 0$ might overestimate the number of large crystals. Conversely, if flocculation dominates, setting $B = 0$ might underestimate the number of large crystals.

**R1:** p. 5, line 32: possibility of confusion due to use of $\delta$ for delta function and thermal boundary layer, possibly resolved by a subscript on thermal boundary layer.

Thanks, we now use $\delta_T$ to denote thermal boundary layer thickness.

**R1:** p. 6, line 1: possibility of confusion due to use of $\epsilon$ for turbulence intensity.

Thanks, we now use $\tilde{\epsilon}$ at this point to denote a small parameter.

**R1:** p. 6, line 15" "calibration"

Changed.

**R1:** p. 7: section 3: Please define precisely what you mean by a "mixed layer" in terms of profiles of temperature, salinity, velocity, ice crystal concentration, etc. How would the presence of waves affect your "mixed layer"?

Please see our earlier comments. Waves might be one source of turbulent mixing, and we now mention this explicitly:

> [...] Such turbulence might be driven, for example in the oceans, by wind, waves, and buoyancy-driven convection. A turbulent flow is mechanically driven in laboratory experiments.

**R1:** p. 9, line 4: "D = 1 m" I assume it is 1 m?

Yes, we added the unit.

**R1:** p. 11, line 13: Contradictory statement that sediment would act as nuclei since you state on p.6, line 5 that Daly expects this to be unlikely.

Yes, Daly does expect this to be unlikely, so we added the comment "perhaps unlikely".

**R1:** p. 11, line 26-31: "experimentalists"? Which experiments are you referring to in this paragraph?

We are now more precise:

[...] which was not controlled in the experiments of Michel (1963) and Carstens (1966) that Svensson and Omstedt (1994) used to test their model

**R1:** p. 12, line 9: 1 m/s is a rather rapid current for the ocean.

We changed the example as follows:

For example, a lateral current of $0.1 \, \mathrm{m/s}$ would take $100 \, \mathrm{s}$ to move material across a lead that is $10 \, \mathrm{m}$ wide.

**R1:** Fig 5 & Fig 6: cropping of symbols on abscissa

The symbols on the abscissa looked okay in our version of the PDF. Could the issue be explained in more detail?

**R1:** p. 13, line 9: why is f2 called the first growth law? Is it not true that f2 $\approx$ 1, rather than $\ll$ 1 (see p.3)

Yes, we removed the word 'first' and now say:

We start with the growth law $f = f_2$ (a constant)

**R1:** Fig 6 caption: by "crystal size" I believe you mean crystal radii?

Yes, we changed "size" to "radii" as suggested. Note that this figure is now numbered Figure 7.

**R1:** p. 14, line 4-5: Please outline how equation (25) is manipulated. I assume equation (24) is substituted into (25) and integrated?

Yes, we now say

Equation (25) can be manipulated by substituting in equation (24) and integrating to show that ...

**R1:** p. 15, line 6: please remind the reader that U indicates turbulence intensity and nmax indicates secondary nucleation.

Thank your for this suggestion. We now say:

Therefore average crystal size depends on (1) secondary nucleation (affected by turbulent intensity through $U_0$ and efficiency of secondary nucleation through $\tilde{n}_{\mathrm{max}}$, where more secondary nucleation means smaller crystals), and on (2) gravitational removal (a larger gravitational removal rate prefactor $\gamma_0$ means larger crystals).

**R1:** p. 15, line 7-13: is it possible to discuss the experimental behavior of secondary nucleation in relation to the model predictions?

We slightly revised our comment about estimating secondary nucleation efficiency as follows:

In geophysical settings and laboratory experiments, the crystal rise velocity, mixed-layer depth and turbulent intensity can be measured much more easily than the efficiency of secondary nucleation. We therefore suggest choosing this parameter to match with observations of average crystal size. For example, choosing the reduced value $\tilde{n}_{\max} = 4 \times 10^5 \text{ m}^{-3}$ would give an average crystal size of about $0.5 \text{ mm}$.

**R1:** p. 16, Table 1: it would be helpful to have the parameters names on the table as a reminder to the reader.

Thank you for this helpful suggestion, we added parameter names to the table.

**R1:** p. 17, Section 4: As stated above I did not understand because it seemed to me that the authors had introduced three growth models, and then did not use any of them in their frazil-laden plume study. Instead they ran sensitivity tests on two "old" models.

Please see our earlier response

**R1:** p. 18, line 16: "narrow slightly"? Do you mean D becomes slightly smaller?

Yes, we now say this explicitly:

have a slightly smaller depth $D$

**R1:** p. 18, line 20: supercooling is "similar" to what?

We changed the sentence order to clarify this:

A faster crystal growth rate is associated with a faster increase in crystal concentration along the slope, although similar quasi-steady states are reached after the supercooling is almost exhausted.

**R1:** p. 18, line 34: "(panel g)"? Do you mean panel c?

Thanks for spotting this, we meant panels c and f, and now say so.

**R1:** p. 19, Fig 8: why is f in the legend since it does not seem to change between model runs? Should it say f1 or f2 or f3?

Thank you for this suggestion. We edited the figure legend (please see revised figure).

**R1:** p. 19, Fig 8: In the caption we are told that calculations with f1 are similar to SO94. But surely it would be more relevant to this paper to show f1 results, rather than a sensitivity study of an "old" model?

Yes, we agree. Please see our earlier comment for the detailed changes we made.

**R1:** p. 18 & 19, Fig 8: I suggest reminding the reader which is the "slow" growth and which is the "fast" growth model by explicitly stating this in the legend. I got rather confused in the discussion on p. 18.

Thank you for this suggestion. We edited the figure legend (please see revised figure).

**Notes:** Blue denotes the referee's comment; black denotes our response; red denotes an extract from the revised manuscript. Please see the 'tracked changes' version of the manuscript for a comprehensive list of changes. Note that figure numbers have changed in revision.

**2    Referee 2 (R2)**

5    **R2:** This is an interesting and well-written paper that discusses the modelling of frazil ice formation in mixed layers at the surface of the ocean and beneath ice shelves. The presentation is generally clear and logically structured, and should be easy to follow for those familiar with earlier literature. If I have one slight criticism, it is that there are a few key papers that form the background to this work which the authors assume the reader will already be familiar with or will read alongside their paper. To a certain extent that is inevitable, and should not really be a problem, but I personally cannot easily access at least one of

10   the key papers that I do not know, and I suspect that many might be in a similar position. While I am not suggesting adding greatly to the size of this paper with lengthy reviews of earlier work, there are a couple of places where I think a few extra details would help. Other than that I found little to fault and would suggest that the paper is acceptable with minor revision along the following lines:

We thank the referee for their very positive assessment of our manuscript. We appreciate their constructive suggestions for

15   minor revisions, and address all points below. We have expanded our discussion of some of the essential background to this paper, and added a new figure, in light of the referee's suggestions.

**R2:** 1) I found the introduction to the crystal growth parameterisations on page 3 (lines 1-20) unnecessarily opaque. I think a few more details of the Rees Jones and Wells (2015) work may have helped. I assume that the discussion of the heat flux around a disc-shaped crystal, and the relevant boundary layer thickness, at the top of page 3, comes from that work, as does the

20   expression for f1 that comes a little later. I think a few details (with maybe a diagram) of the temperature distribution around a growing disc, to give some insight into where f1 comes from would be really informative. Also it would seem more logical to discuss this first, before the approximations, especially if it is to be denoted by the number 1. (Later on page 13, line 9, f2 is somewhat confusingly referred to as the "first" growth law, presumably because of this slightly illogical sequencing that sees f1 introduced last.) With the temperature distribution around a growing disc described and the correct boundary layer scaling

25   justified, the approximations f2 and f3 can then be put in a better context. Don't they come from the assumption of a spherically symmetric temperature distribution, around the disc edge in the case of f2 and around the entire disc in the case of f3?

Thank you for these comments and for the very helpful suggestion of adding a new diagram. We added a new Figure 1 that gives an example of the temperature distribution around a disk-shaped crystal. We refer to this figure when explaining the boundary layer thickness scaling arguments, which we hope will make the discussion easier to follow. We also reordered the

30   discussion of the three parameterizations in line with the referee's suggestion. The new figure and reordered paragraph can be found in the tracked-changes document (section 1.2). We also report the key revised sections and new figure below:

[Figure]

**Figure R1.** Example of temperature distribution around a disk-shaped crystal (filled grey region outlined in black) of radius 1 mm and thickness 0.1 mm. Contours of temperature are shown varying between the freezing temperature $T_f$ and the far-field temperature $T_0 < T_f$. In this example the crystal is growing into freshwater, and the thermal conductivity of ice is four times larger than than of water. The numerical calculations used to make this figure are described in Rees Jones and Wells (2015).

In general, the radial growth rate can be written in the form

$$\rho_i L G = (\mathrm{Nu}\, k_l \Delta T / H) f, \tag{1}$$

where $\rho_i$ is the density of ice, $L$ is the latent heat of solidification, Nu is the crystal Nusselt number which equals 1 for purely diffusive growth and can be enhanced by flow, $k_l$ is the thermal conductivity of the liquid phase, $\Delta T$ is the amount of supercooling below the in-situ freezing temperature, and $f$ is a dimensionless geometric factor. A helpful way to interpret equation (1) is to rearrange it into an expression for the rate of crystal-mass growth, with the ice-crystal mass $M = \rho_i \pi R^2 H$. We find

$$L\frac{dM}{dt} = \mathrm{Nu} k_l \Delta T 2\pi R f \propto \mathrm{Nu} k_l \Delta T \frac{A}{\delta_T} f, \tag{2}$$

The right-hand side is the product of the area for heat transfer $A$ and the heat flux scale $k_l \Delta T / \delta_T$, where $\delta_T$ is a thermal boundary layer thickness. Numerical calculations of the temperature distribution around an ice crystal (an example is shown in figure 1) show that $\delta_T \propto H$ near the crystal edges, which have an area $A \propto RH$. However, $\delta_T \propto R$ near the crystal faces, which have an area $A \propto R^2$. In either case, the ratio $A/\delta_T \propto R$. Thus the scaling argument suggests $f \propto 1$ (*cf.* equation 2). It is interesting to note that the mass growth rate of spherical crystals is also proportional to crystal radius $R$, so the rate of latent heat release seems to depend on crystal size $R$ but not on the details of the geometry.

We now consider three possible parameterizations of crystal growth, which we denote $f_{1,2,3}$. Numerical calculations of the heat transfer by diffusion from a disk-shaped crystal (Rees Jones and Wells, 2015) show that the growth

rate depends logarithmically on aspect ratio $f_1(h = H/2R) = 1/[0.9008 - 0.2634\log(h)]$, which is similar to 1. Some previous studies [...]

**R2:** 2) Sections 3 and 4 discuss applications to two geophysical situations, where an assumption has been implicitly made that all properties are either well-mixed or follow some simple self-similar shape, so that depth-integration produces simple depth-averaged properties (and products of properties). While such an assumption is quite common, it is more questionable in this case than is usual. The term $\Delta T$ in equation (14) has a pre-defined depth-dependence. Even if the layer is uniform in temperature, the super-cooling will be a linear function of depth within the layer, because the freezing temperature is pressure-dependent. Furthermore, the ice concentration cannot be well-mixed, because the distribution of crystals will be determined not just by turbulent diffusion, but also by their buoyant rise. So the concentration will be highest at the top of the layer where the super-cooling will also be a maximum. The use of depth-mean quantities is common in the literature, but I think it would be worthwhile to point out the limitations of the assumption and the potential impact on the results. For example, could the finding that increasing D promotes "frazil explosions" (page 8, lines 22-23 and figure 3b) be an artefact of this assumption?

We thank the referee for raising these important issues relating to the assumption of a well mixed layer. Firstly, the freezing temperature depends on depth, so $\Delta T$ has a depth-dependence. This depth-dependence is significant when the mixed layer is sufficiently deep that the freezing point change is comparable to the calculated supercooling. In laboratory settings the effect is always small, but in deep ocean mixed layers of $O(100\text{ m})$ the effect is noticeable. We added a sentence explaining this limitation at the start of section 3.

Note that, in this section, we neglect the depth-dependence of the freezing temperature, which affects the supercooling $\Delta T$. This is a good approximation provided the mixed layer is relatively shallow, but would not be appropriate for mixed layers deeper than $O(100\text{ m})$.

Secondly, crystal concentration will have a depth-dependence. This is also a good point, although we don't think that our finding that increasing $D$ increases the likelihood of a frazil explosion is an artefact of the assumption. We added a couple of sentences to address the issue:

$\gamma = W/D$ is an effective gravitational removal term. In reality, crystal concentration would tend to decrease with depth (Svensson and Omstedt, 1998) because of crystal buoyancy. Nevertheless, $\gamma = W/D$ is an appropriate scaling relationship because removal increases with crystal rise velocity $W$ and decreases with mixed layer depth $D$ because turbulent eddies act to mix crystals down to that depth range. This type of depth-integrated representation of the process of gravitational removal has been used successfully in previous studies of turbulent, particle-laden gravity currents (Bonnecaze et al., 1993).

**R2:** 3) In section 4 a few words about the plume model behaviour might help to put the results presented in figure 8 into context. Referring the reader to earlier publications for much of the detail is fine, but introducing the basic concept of a buoyant

flow generated by melting that subsequently becomes supercooled because of the fall in pressure as it ascends the ice shelf base would be helpful. Also it would be worthwhile pointing out that, since the plume flows along an ice-ocean boundary, it drives direct freezing onto the boundary as well as growth of frazil crystals that can be deposited if the plume flow is weak enough. This would not require many extra words but would clarify for the reader new to the concepts what the three panels on the right-hand side of figure 8 actually show.

This is an excellent idea, and we added a few extra sentences to explain the basic physical concepts. We also now mention direct basal freezing. The revised sentences read:

> Frazil ice also forms in plumes of ice shelf water (ISW) beneath floating ice shelves. A plume is fed by the discharge of subglacial meltwater at the start of the shelf and by melting from the shelf itself. These meltwaters are relatively fresh, so the plume rises buoyantly. The plume entrains ocean waters, resulting in an intermediate temperature and salinity called ice shelf water (ISW). [...] The plume becomes supercooled as it ascends the ice shelf base because of the fall in pressure and consequent change in the freezing temperature. This supercooling leads to a combination of frazil-ice formation and direct basal freezing. Frazil ice increases the plume buoyancy and so accelerates the plume.

**R2:** 4) Finally, some more minor comments:

Page 1, line 17: ". . . phase of ice growth in turbulent waters."

Corrected.

**R2:** Page 1, line 20: ". . . occurs when it is cooled efficiently . . ."

Corrected.

**R2:** Page 2, line 6-7: Actually Engelhardt and Determann (1987) did not drill a borehole through the ice, nor did they observe the granular texture of the ice. They used a hotwater drill, so could not recover any samples, although did infer that the bottom 35 m of the ice shelf consisted of a slushy layer of unconsolidated frazil ice. The ice at the bottom of Ronne Ice Shelf was not sampled until a little later (Oerter et al., 1992, Nature, 358, 399-401).

Thank you for this clarification. The revised sentence now reads:

> Some of the ice precipitates onto the base of the ice shelf, where it forms so-called marine ice, which has a granular texture. The presence of marine ice was inferred and subsequently observed by drilling boreholes through the ice shelf (Engelhardt and Determann, 1987; Oerter et al., 1992).

**R2:** Page 2, line 31: ". . . crystal-mass growth, with the ice crystal . . ."

Corrected.

**R2:** Page 12, line 6: ". . . transient differences are therefore . . ."

Corrected.

**R2:** Page 13, line 6: " . . . order to understand better the physical . . ."

Corrected

**R2:** Page 17, line 9: See earlier comments (2). You mean when the plume becomes supercooled in a depth-averaged sense?

In this introductory paragraph we are describing the general geophysical problem, before we have introduced a depth-averaged model, so we didn't change this phrase. In terms of the model described in the subsequent paragraph, yes, frazil-ice grows when the plume becomes supercooled in a depth-averaged sense. We added the sentence:

Note that we also average the freezing temperature over the depth of the plume.

**R2:** Page 17, line 20-23: Because you only discuss freezing, this comment seems a little out of place. Don't Holland and Jenkins (1999) set the conductive flux to zero when there is freezing at the ice shelf base? Nevertheless, this comment might fit into a slightly expanded description of the plume (see earlier comments (3)).

This technical aside resulted from discussion with Adrian Jenkins about the model configuration used in Smedsrud and Jenkins (2004), which we tried to mimic as closely as possible. Note that there can be both basal freezing and melting (melting occurs nearer the grounding line, as shown by Jenkins and Bombosch, 1995, figure 5d). In our figure, we only plot the region where frazil-ice forms. We slightly expanded our comment, which now reads:

Note that our thermal calculation includes an estimate of the conductive heat flux into the ice shelf (based on the thermal boundary layer parameterization of Holland and Jenkins (1999), using a core ice-shelf temperature of $-15\,°C$, A. Jenkins, personal communication).

**R2:** Page 17, line 32: ". . . over O(100 km) and the plume is . . ."

Thanks, we corrected this.

**R2:** Page 18, line 34: In the parentheses, should that be "(panel f)"?

Thanks for spotting this, we meant panels c and f, and now say so.

**R2:** Page 18, first paragraph and caption to figure 8: By "fast" and "slow" growth, do you mean f2 and f3? Perhaps you could clarify the point.

Yes, we edited the text, figure legend and caption in response to this comment, and those of the other referee, as follows. In the first paragraph of section 4.1, our revised text now reads:

[revised manuscript text omitted]